# Impact of the 3% Oxygen Desaturation Index via Overnight Pulse Oximetry on Cardiovascular Events and Death in Patients Undergoing Hemodialysis: A Retrospective Cohort Study

**DOI:** 10.3390/jcm12030858

**Published:** 2023-01-20

**Authors:** Yasuhiro Mochida, Takayasu Ohtake, Kunihiro Ishioka, Machiko Oka, Kyoko Maesato, Hidekazu Moriya, Sumi Hidaka, Shuzo Kobayashi

**Affiliations:** Kidney Disease and Transplant Center, Shonan Kamakura General Hospital, Kamakura 247-8533, Kanagawa, Japan

**Keywords:** sleep-disordered breathing, hemodialysis, cardiovascular event, mortality

## Abstract

It is unclear whether the severity of sleep-disordered breathing (SDB) affects the risk of cardiovascular events and mortality in patients undergoing hemodialysis (HD). We determined the severity of SDB with the 3% oxygen desaturation index (ODI) via overnight pulse oximetry. This study was a retrospective cohort, observational study of 134 patients on maintenance HD at a single center. They were divided into four groups according to SDB severity (normal, mild, moderate, and severe), and were followed. The baseline characteristics of all patients were as follows: the median age was 67 (interquartile range, 59–75) years, 64.2% were men, 37.3% were diabetic, and the median duration of HD was 69 (29–132) months. During follow-up, major adverse cardiovascular events (MACEs) occurred in 71 patients and deaths in 60 (including 32 cardiovascular deaths). Severe SDB was an independent risk factor for MACEs (hazard ratio [HR] = 4.66, 95% confidence interval [CI] = 1.87–11.61, *p* = 0.001) and all-cause death (HR = 5.74, 95% CI = 1.92–16.70, *p* = 0.001). Severe SDB had a statistically significant impact on the risk of MACEs and mortality in patients undergoing HD. The severity of the 3% ODI via overnight pulse oximetry may be a useful marker as a risk factor for cardiovascular outcomes and mortality in these patients.

## 1. Introduction

Sleep-disordered breathing (SDB) is a chronic disorder caused by obstruction of the upper airway during sleep. SDB reportedly causes nocturnal hypoxia and nocturnal sympathetic nerve activation [1], elevates blood pressure [2,3], and contributes to cardiovascular events (stroke [4,5,6], heart failure [7,8,9], arrhythmia [5], and death [4,10,11]) in the general population.

The reported prevalence of SDB in the general middle-aged population is 2 to 4% [12]. However, SDB is even more prevalent (32–82%) [13,14,15,16] in patients undergoing hemodialysis (HD), with 26% of such patients reportedly having severe SDB [17]. Risk factors for SDB in the general population, such as older age, male sex, obesity, smoking, a large neck circumference, and diabetes, [18] are also prevalent in the population with chronic kidney disease. However, the impact of the severity of SDB on future macrovascular cardiovascular events (cardiac events, cerebrovascular events, and events related to peripheral artery disease [PAD]) and cardiovascular mortality in patients undergoing HD is unclear.

The apnea-hypopnea index (AHI), evaluated with polysomnography (PSG), is the gold standard for the determination of the severity of SDB [19]. However, PSG is a complicated procedure that requires hospitalization. SDB severity can also be determined with the 3% oxygen desaturation index (ODI), evaluated via nocturnal pulse oximetry, which is simpler to perform and does not require hospitalization.

The aim of our study was to investigate the prognosis of patients undergoing HD, in terms of cardiovascular events and mortality, according to SDB severity determined by the 3% ODI with overnight pulse oximetry.

## 2. Materials and Methods

### 2.1. Study Design and Participants

This was a retrospective cohort, observational study. A total of 167 patients undergoing maintenance HD were recruited in Shonan Kamakura General Hospital from December 2012 to April 2013. The inclusion criteria for this study were patients aged from 18 to 85 years, and an HD duration of ≥3 months. The exclusion criteria were as follows: having pulmonary disease, undergoing combined peritoneal dialysis and HD, undergoing chemotherapy for malignancy, having undergone renal transplantation within the previous 6 months, being treated for SDB with devices (a mouthpiece and a continuous positive airway pressure machine, etc.) during the follow-up period, and being unable to cooperate or to provide consent in evaluating overnight pulse oximetry. Among the 167 patients, 18 did not provide informed consent, 5 were >85 years old, 5 had an HD duration of <3 months, 1 had idiopathic interstitial pneumonia, 3 had undergone renal transplantation, and 1 was treated for SDB with a continuous positive airway pressure device. Finally, 134 patients who evaluated SDB were enrolled and followed cardiovascular events and mortality (Figure 1).

### 2.2. Measurements and Definitions

After evaluating overnight pulse oximetry, these patients were followed up for about 8 years (up to and including April 2021). SDB was defined as five or more 3% ODI events per hour via overnight pulse oximetry. Patients were divided into four groups according to SDB severity (normal, <5 events of 3% ODI; mild, ≥5 and <15 events; moderate, ≥15 and <30 events; and severe, ≥30 events). Major adverse cardiovascular events (MACEs) included cardiac events, cerebrovascular events, PAD events, and cardiovascular death. Cardiac events were defined as congestive heart failure requiring hospitalization, ischemic heart disease (angina or myocardial infarction that was confirmed with an electrocardiogram, and coronary stenosis or obstruction that was confirmed with percutaneous transluminal coronary arteriography and accompanied by symptoms of angina), symptomatic aortic stenosis, and sudden death. Cerebrovascular events were defined as strokes (cerebral infarction and hemorrhage) and transient ischemic attacks (TIAs). PAD events were defined as critical limb ischemia or percutaneous transluminal arteriographic characteristics indicative of femoral or lower-limb artery stenosis. Cardiovascular deaths were defined as deaths related to heart disease, stroke, and PAD, including sepsis and ulcers due to severe critical limb ischemia.

### 2.3. ODI

The ODI was investigated via pulse oximetry using a pulse watch (PMP-200GplusX, PHILIPS, Tokyo, Japan), which measures arterial oxygen saturation of the forefinger. The patients underwent pulse oximetry with the contralateral arm to the vascular access through the night after they underwent HD. After the pulse watch was removed, the recorded data were analyzed for 3% ODI events via device-specific analysis software. The 3% ODI referred to the number of events per hour in which oxygen saturation decreased by ≥3% from baseline.

### 2.4. Patient and Data Collections

The following patient characteristics were obtained from the electronic records of our hospital: age, sex, HD duration (months), body mass index (BMI), cause of end-stage renal disease, history of diabetes mellitus (DM), angina or myocardial infarction, stroke or TIAs, and PAD. Systolic and diastolic blood pressures were measured in the semi-supine position 10 min before starting HD. Blood samples were obtained before starting HD on the first day of dialysis, for blood counts and various biochemical tests. Echocardiograms were used to measure the ejection fraction and left ventricular mass index (LVMI). We evaluated these patients’ electronic hospital records for cardiovascular events and death, as well as the development of new-onset atrial fibrillation (AF).

### 2.5. Study Endpoints

The primary endpoints were the cumulative incidences of MACEs and all-cause mortality according to SDB severity. The secondary endpoints were the cumulative incidences of cardiac, cerebrovascular, and PAD events, as well as AF, according to SDB severity.

### 2.6. Statistical Analysis

Nominal variables are expressed as frequencies and percentages. Continuous variables were presented as medians (interquartile ranges [IQRs]). Patient characteristics were analyzed between groups with the Cochran–Armitage test and the Jonckheere–Terpstra test. The cumulative probability of incidence of MACEs and all-cause mortality were calculated with the Kaplan–Meier method and comparisons of these between the SDB-severity groups were made by using the Wilcoxon–Breslow–Gehan test for trend. Multivariable analysis was performed by using Cox proportional-hazards regression to examine the association between SDB severity and events (MACEs and all-cause death). The final multivariable model was used to control for potential confounding factors of age, sex male, BMI, duration of HD, the existence of DM, and CRP. The cumulative probability of incidences of cardiac, cerebrovascular, and PAD events, as well as the new incidence of AF, according to SDB severity was evaluated with Wilcoxon–Breslow–Gehan test for trend. All statistical analyses were performed using Stata software version 17 (StataCorp LLC, College Station, TX, USA). A *p*-value of < 0.05 was considered statistically significant.

## 3. Results

### 3.1. Patient Characteristics

A total of 134 patients with HD were evaluated. Table 1 summarizes the baseline patient characteristics. The median age was 67 years, 64.2% of the patients were men, and 37.3% had DM. The causes of end-stage renal disease were included DM in 43 patients, chronic glomelular-nephritisits (CGN) in 32, polycystic kidney disease (PKD) in 11, nephro-sclerosis in 10, and the others in 38. All of these patients underwent hemodialysis with arteriovenous fistula or graft three times a week. SDB severity distribution of the 134 patients was as follows: normal = 22 (16.4%), mild = 52 (38.8%), moderate = 37 (27.6%), and severe = 23 (17.2%) (Figure 1). Median oxygen saturation levels (saturation of percutaneous oxygen, SpO_2_,) at rest with the device attached were 98% and not significant. The median number of 3% ODI events per hour in each of these groups was 3.7, 8.6, 21.0, and 48.2, respectively. The severity of ODI was associated with men (*p* for trend < 0.001), BMI (*p* for trend = 0.021), duration of HD (*p* for trend = 0.019), and CRP (*p* for trend = 0.010).

### 3.2. Primary Endpoints

The median observational period was 37 (IQR: 18–87) months. During the follow-up period, MACEs occurred in 71 patients, cardiovascular death in 32 patients, and all-cause mortality in 60 patients. The causes of death included cardiac disease in 20 patients (ischemic heart disease in 10, symptomatic cardiac disease due to aortic valve stenosis in 5, sudden death of unknown cause in 5), stroke in 6 (cerebral infarction in 3, cerebral hemorrhage in 3), PAD in 6 (all 6 patients had sepsis, 4 of whom died suddenly), non-PAD-related infections in 11 (pneumonia in 5, sepsis in 2, catheter-related infection in 1, other infections in 3), gastrointestinal bleeding in 2, malignancy in 10, and other causes in 5. Figure 2A illustrates the cumulative probability of incidence of MACEs, and Figure 2B illustrates the cumulative probability of incidence of all-cause death, both according to SDB severity. The trend test using Wilcoxon–Breslow–Gehan test for the Kaplan–Meier curves revealed that SDB severity is significantly associated with the frequency of MACEs (*p* for trend = 0.0015) and all-cause death (*p* for trend = 0.026).

Cox proportional-hazards regression analysis revealed that, compared with normal SDB, severe SDB independently contributed to MACEs ([HR] = 4.66, 95% confidence interval [CI] = 1.87–11.61, *p* =0.001)) (Table 2A) and all-cause death (HR = 5.74, 95% CI = 1.92–16.70, *p* = 0.001) (Table 2B). Likewise, sensitivity analyses with adjustment for factors of differences in SDB groups (i.e., men, BMI, duration of HD, and CRP) resulted in similar findings.

### 3.3. Secondary Endpoints

Cardiac events occurred in 49 patients, including 27 with ischemic heart disease, 8 with congestive heart failure requiring hospitalization, 9 with heart disease due to aortic stenosis, and 5 with a sudden death of unknown cause. Cerebrovascular events occurred in 32 patients, of whom 25 had cerebral infarction, 1 had a TIA, and 6 had cerebral hemorrhage. PAD events occurred in 34 patients. New-onset AF events occurred in 18 patients. We compared these events between SDB severity groups (Appendix A). The incidence of cardiac events was evaluated with the Wilcoxon test for trend. The result revealed that the severity of SDB is associated with cardiac events (*p* for trend = 0.026). The incidence of cerebrovascular events (*p* for trend = 0.12) and PAD (*p* for trend = 0.08) was not associated with the severity of ODI. The new incidence of AF was also higher in patients with severe SDB (*p* for trend = 0.016, Appendix A). A total of 15.4% (6/39) of the patients with new-onset or a history of AF had a stroke or TIAs during the observational period.

## 4. Discussion

In this study, we aimed to evaluate whether the severity of SDB, measured by using nocturnal pulse oximetry (which is much simpler to perform than PSG), can have an impact on the combination of cardiovascular events and all-cause mortality in patients undergoing HD. We discovered that severe SDB of 3% ODI via overnight pulse oximetry is an independent risk factor for both cardiovascular events and all-cause mortality in such patients and that severity of 3% ODI may be a useful marker as a risk factor for cardiovascular events.

Several cross-sectional studies have revealed that the ODI, assessed via pulse oximetry, is associated with cardiovascular risk factors [20]. However, there have been few cohort studies in which the validity of the ODI as a prognostic factor was determined. In one of those, Masuda et al. [21] reported that SDB, measured with pulse oximetry, was an independent risk factor and prognostic parameter for cardiovascular events and mortality in a group of 94 patients; our results corroborate theirs, although they did not evaluate the effect of different degrees of SDB severity. Indeed, one of the limitations of their study was that there were fewer patients with severe SDB than in other studies [20,22,23]. In the present study, as in previous cohort studies, many patients with moderate-to-severe SDB were included, and these patients were followed up for approximately 8 years with overnight pulse oximetry following HD. Another study revealed that nocturnal mean oxygen saturation, rather than ODI, was associated with all-cause mortality [24]. However, in that study, a 4% threshold was used for ODI, along with the AHI, and the authors included patients with stage five chronic kidney disease before the initiation of maintenance HD, who might have had excessive fluid; moreover, a large proportion of the patients were transplant recipients (47%), >20% of whom died during the follow-up period. The differences in patient backgrounds between our study and theirs may explain the difference in risk factors. The novelty of our study included the analysis of cardiovascular composite events, all-cause mortality, and each cardiovascular event (cardiac, cerebrovascular, PAD, and new-onset AF), as well as the inclusion only of patients undergoing maintenance HD. We demonstrated that severe SDB was associated with the combination of cardiovascular events (especially the onset of cardiac events), as well as with all-cause mortality compared to normal SDB.

Instead of evaluating AHI with PSG, we evaluated the 3% ODI with overnight pulse oximetry to assess the severity of SDB, and we discovered that this metric was useful in predicting the prognosis of patients undergoing HD in terms of cardiovascular events and all-cause mortality. PSG is a widely-used tool for the assessment of obstructive sleep apnea [19] and has been used in combination with many techniques (electro-encephalography, cardiography, and myography of the lower extremities, respiratory inductance plethysmography, nasal pressure transduction, pulse oximetry of the fingers, and video monitoring) to obtain detailed information such as the AHI, apnea type (obstructive, central, or mixed), nocturnal arterial blood oxygen saturation, sleeping type (rapid eye movement [REM] or non-REM), and leg movement. Therefore, the AHI, obtained with PSG, is used as a standard diagnostic and prognostic parameter for SDB. However, a large number of patients undergoing HD (32–82%) [13,14,15,16] have SDB, and few of the hospitals that these patients attend are equipped with facilities for PSG. On the other hand, as pulse oximetry makes use of a portable device, it can be used to measure arterial blood oxygen saturation at home via the index finger. Masuda et al. [25] and Jung et al. [20] previously demonstrated that 3% ODI assessed via pulse oximetry was statistically significantly correlated with AHI assessed via PSG. Hence, SDB via overnight pulse oximetry may be a useful marker as a risk factor for cardiovascular events and all-cause mortality.

The new incidence of AF tended to be higher as the severity of SDB increased, and 15.4% of patients with AF experienced cerebral infarction and/or transient ischemia. In a previous report, the presence of SDB in patients undergoing HD was associated with new-onset AF [26], and the onset of AF in patients undergoing HD was associated with the risk of stroke [27]. On the other hand, Mitsuma et al. [28] reported that AF was not statistically significantly associated with the development of cerebral infarction, but that it was with all-cause and cardiovascular mortality. In light of this study, it may be necessary to evaluate the SDB severity in patients undergoing maintenance HD who develop AF and to monitor them closely for the onset of cardiovascular events, even though the incidence of stroke in such patients is not high.

Our study had several limitations. First, we did not investigate the AHI with PSG. Although the 3% ODI does not allow the determination of whether a patient has obstructive apnea or central apnea, SDB measured with the AHI and that measured with the ODI are reportedly correlated [25]; hence, we believe that the pulse oximetry used in this study was suitable for the determination of the severity of sleep apnea. Second, patients were evaluated using overnight pulse oximetry only one time. Third, the sample size of this study was small. Therefore, it is possible that not all confounding factors have been eliminated. Finally, this was an observational rather than an interventional study. Therefore, we do not know whether the resolution of sleep apnea, such as with continuous positive airway pressure treatment, reduces the risk of cardiovascular events.

## 5. Conclusions

Severe SDB, as evaluated using 3% ODI via overnight pulse oximetry, had a statistically significant impact on cardiovascular events and all-cause mortality in patients undergoing HD. The severity of 3% ODI via overnight pulse oximetry may be a useful marker as a risk factor for cardiovascular outcomes and mortality in such patients. Future studies with larger sample sizes are needed to determine whether SDB treatments, such as continuous positive airway pressure, can improve cardiovascular outcomes in patients undergoing HD.

## Figures and Tables

**Figure 1 jcm-12-00858-f001:**
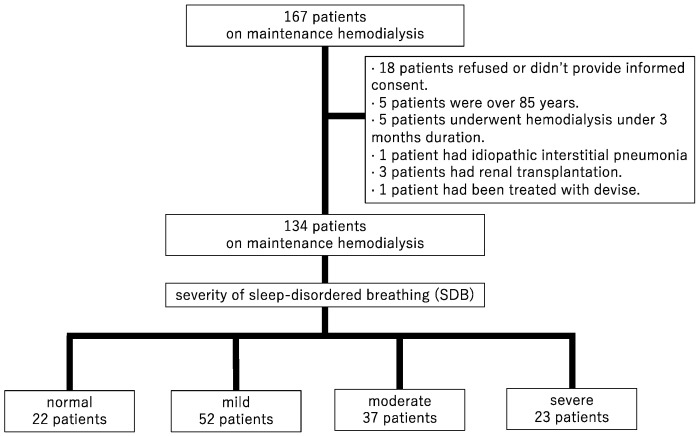
Patient selection flow chart.

**Figure 2 jcm-12-00858-f002:**
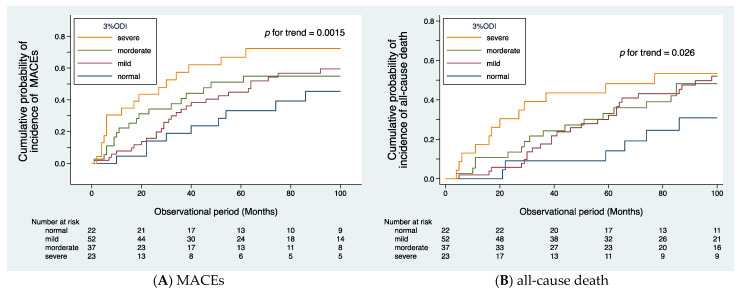
Cumulative probability of incidence of cardiovascular events (**A**) and all-cause death (**B**) according to the severity of SDB. Kaplan–Meier analysis revealed that the patients in the severe group had a higher frequency of cardiovascular events (*p* for trend = 0.0015, (**A**)) and all-cause death (*p* for trend = 0.026, (**B**)).

**Table 1 jcm-12-00858-t001:** Baseline characteristics of patients according to SDB severity.

	Total	Normal	Mild	Moderate	Severe	*p* for Trend
	N = 134	*n* = 22	*n* = 52	*n* = 37	*n* = 23	
3% ODI, times/h	11.3 (6.2–25.5)	3.7 (1.7–4.7)	8.6 (6.4–10.2)	21.0 (19.0–25.8)	48.2 (35.8–56.5)	<0.001
SpO_2_ at rest, %	98 (97–99)	98 (98–99)	98 (98–99)	98 (97–98)	98 (96–99)	0.065
Age, y	67 (59–75)	64 (61–72)	71 (59–75)	70 (63–76)	67 (59–75)	0.50
Men, %	86 (64.2%)	11 (50.0%)	25 (48.1%)	31 (83.8%)	19 (82.6%)	<0.001
BMI, kg/m^2^	20.5 (18.5–23.0)	18.9 (17.8–21.0)	19.8 (18.5–22.3)	20.9 (19.4–23.9)	22.5 (20.5–25.1)	0.021
Duration of HD, months	69 (29–132)	120 (43–236)	83 (31–132)	49 (13–115)	61 (35–105)	0.019
Vascular access						
AVF	124 (92.5%)	22 (100.0%)	45 (86.5%)	34 (91.9%)	23 (100.0%)	0.095
AVG	10 (7.5%)	0 (0.0%)	7 (13.5%)	3 (8.1%)	0 (0.0%)	
CVC	0 (0.0%)	0 (0.0%)	0 (0.0%)	0 (0.0%)	0 (0.0%)	
DM, %	50 (37.3%)	8 (36.4%)	23 (44.2%)	13 (35.1%)	6 (26.1%)	0.29
History of CVD, %	63 (47.0%)	8 (36.4%)	25 (48.1%)	15 (40.5%)	15 (65.2%)	0.17
History of IHD, %	31 (23.1%)	1 (4.5%)	16 (30.8%)	9 (24.3%)	5 (21.7%)	0.42
History of CBVD, %	24 (17.9%)	4 (18.2%)	6 (11.5%)	5 (13.5%)	9 (39.1%)	0.079
History of PAD, %	23 (17.2%)	1 (4.5%)	10 (19.2%)	6 (16.2%)	6 (26.1%)	0.14
AF, %	21 (15.7%)	2 (9.1%)	9 (17.3%)	4 (10.8%)	6 (26.1%)	0.33
SBP, mmHg	150 (130–160)	150 (140–160)	150 (130–160)	150 (130–160)	150 (130–160)	0.69
DBP, mmHg	80 (70–85)	80 (65–85)	75 (70–85)	80 (70–80)	85 (70–90)	0.35
Hb, g/dL	11.3 (10.4–12.2)	11.4 (10.8–11.9)	11.3 (10.3–11.9)	11.3 (10.8–12.8)	11.5 (10.2–12.4)	0.57
Alb, g/dL	3.7 (3.5–3.9)	3.8 (3.6–3.9)	3.7 (3.5–3.9)	3.8 (3.5–3.9)	3.7 (3.5–4)	0.94
P, mg/dL	5.3 (4.6–6.1)	5.1 (4.3–6.4)	5.5 (4.6–6.2)	5.1 (4.7–5.9)	5.4 (4.7–6.9)	0.52
PTH, pg/mL	160 (104–285)	191 (105–299)	159 (108–273)	144 (80–255)	236 (120–378)	0.77
HDL, mg/dL	48 (40–59)	57 (50–64)	46 (38–55)	49 (39–59)	46 (38–61)	0.34
LDL, mg/dL	77 (59–94)	73 (54–79)	77 (65–94.5)	83 (57–96)	74 (63–120)	0.16
GA, %	15.2 (13.5–17.7)	15.2 (13.4–17.9)	16.1 (14.6–18.3)	14.6 (13.1–17)	14.9 (12.4–17.6)	0.12
CRP (mg/dL)	0.10 (0.04–0.23)	0.05 (0.03–0.09)	0.11 (0.05–0.22)	0.13 (0.07–0.41)	0.16 (0.05–0.23)	0.010
LVMI, g/m^2^	127 (104–146)	138 (117–161)	123 (103–139)	120 (101–149)	127 (106–142)	0.29
EF, %	63 (60–67)	65 (62–67)	63 (60–66.5)	63 (60–66)	63 (53–68)	0.20
E/e’	14.4 (11.7–18.8)	14.9 (12.6–19.9)	14.1 (11.8–17.9)	13.4 (11.2–17.5)	15.2 (11.3–19.7)	0.67

ODI, oxygen desaturation index; SpO_2_, saturation of percutaneous oxygen; BMI, body mass index; HD, hemodialysis; DM, diabetes mellitus; AVF, arteriovenous fistula; AVG, arteriovenous graft; CVC, central venous catheter; CVD, cardiovascular disease; IHD, ischemic heart disease; CBVD, cerebrovascular disease; PAD, peripheral artery disease; AF, atrial fibrillation; SBP, systolic blood pressure; DBP, diastolic blood pressure; Hb, hemoglobin; Alb, serum albumin; P, serum phosphate; PTH, parathyroid hormone; HDL, high-density lipoprotein; LDL, low-density lipoprotein; GA, glycoalbumin; CRP, C reactive protein; LVMI, left ventricular mass index; EF, ejection fraction.

**Table 2 jcm-12-00858-t002:** Multivariable Cox regression analysis for (A) MACEs and (B) all-cause mortality according to SDB severity.

(A) MACEs						
	Model 1			Model 2		
	HR	95% CI	*p*	HR	95% CI	*p*
SDB severity						
Normal	ref			ref		
Mild	1.63	0.76–3.47	0.207	1.41	0.66–3.05	0.369
Moderate	1.81	0.79–4.17	0.162	1.75	0.74–4.13	0.199
Severe	3.91	1.61–9.47	0.003	4.66	1.87–11.61	0.001
Age, ×10 year	1.25	0.96–1.61	0.10	1.25	0.96–1.64	0.090
Men	1.29	0.76–2.19	0.339	1.15	0.69–1.94	0.608
BMI, kg/m^2^	0.91	0.84–0.99	0.003	0.89	0.82–0.97	0.009
Duration of HD, year				1.00	0.99–1.00	0.050
DM				1.80	1.08–2.99	0.023
CRP, mg/dL				0.92	0.61–1.37	0.685
**(B)** **All-Cause Mortality**						
	**Model 1**			**Model 2**		
	**HR**	**95% CI**	** *p* **	**HR**	**95% CI**	** *p* **
SDB severity						
Normal	ref	-	-	ref	-	-
Mild	2.25	0.92–5.52	0.077	2.05	0.83–5.07	0.119
Moderate	1.90	0.73–4.97	0.187	1.70	0.62–4.47	0.307
Severe	4.75	1.69–13.32	0.003	5.74	1.92–16.70	0.001
Age, ×10 year	1.81	1.33–2.47	<0.001	1.77	1.30–2.41	<0.001
Men	1.36	0.77–2.47	0.285	1.11	0.61–2.01	0.741
BMI, kg/m^2^	0.85	0.77–0.94	0.002	0.84	0.75–0.93	0.001
duration of HD, year				0.97	0.93–1.01	0.118
DM				1.20	0.87–2.69	0.139
CRP, mg/dL				1.33	1.04–1.62	0.020

SDB, sleep-disordered breathing; HR, hazard ratio; CI, confidence interval; BMI, body mass index; HD, hemodialysis; DM, diabetes mellitus; CRP, C-reactive protein.

## Data Availability

Data used in this study are available on request to the corresponding authors.

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
