# Peer review of "Impact of the 3% Oxygen Desaturation Index via Overnight Pulse Oximetry on Cardiovascular Events and Death in Patients Undergoing Hemodialysis: A Retrospective Cohort Study"

_jcm, 2023, doi:10.3390/jcm12030858_

Round 1
Reviewer 1 Report
The article is very interesting and the information may be very useful.
I have just few points to be clarify
1) To me, It is not clear the study design. Is retrospective or prospective study? The author declared that the study is retrospective but in the material 2.2 they wrote that they divided the patients according SDB severity and followed up for 8 years. I understood that authors performed ODI, than they divided the patient according to the SDB severity and followed up them for 8 years to evaluate the incidence of MACE in the 4 groups. Is it right? The authors have to clarify this point.
2) The temporal starting point of the study has to be clarify. Did the follow-up start after the ODI evaluation?
3) Why the patients had the ODI done? For the study o for routine clinical evaluation?
4) Is the ODI evaluated only one time for patients? Can the authors support (maybe with some literature) the fact that one ODI evaluation is enough to categorize patients?
5) Is the ODI evaluated in all the patients during the same inter-dialytic interval (long or short)?
6) It is useful to show the oxygen values measured during the ODI evaluation in the population and in the 4 subgroups.
7) The patients’ vascular access type (CVC vs AVF) is an important information to insert in the table.
Author Response
Reviewer1 COMMENTS
The article is very interesting and the information may be very useful.
I have just few points to be clarify
1) To me, It is not clear the study design. Is retrospective or prospective study? The author declared that the study is retrospective but in the material 2.2 they wrote that they divided the patients according SDB severity and followed up for 8 years. I understood that authors performed ODI, than they divided the patient according to the SDB severity and followed up them for 8 years to evaluate the incidence of MACE in the 4 groups. Is it right? The authors have to clarify this point.
→Thank you for pointing out an important issue. In the past, the primary objective was to determine and evaluate the morbidity of sleep apnea syndrome in dialysis patients at our institution. The current study is an attempt to determine the prognosis of these patients and was not originally planned for this design in this investigation of overnight pulse oximetry. As the Reviewer commented, we divided the patients, according to the SDB severity after investigating overnight pulse oximetry previously performed and followed the patients over time to a point in pastime prospectively. Therefore, this study was a retrospective cohort observational study. I changed the Abstract on P1L13, Methods P2L51, and P2L63-64 in red letters.
2) The temporal starting point of the study has to be clarified. Did the follow-up start after the ODI evaluation?
→Thank you for pointing out the issue. The study defines the date of the overnight pulse oximetry test as the beginning of the observation period. I added " After evaluating overnight pulse oximetry, these patients were followed up for about eight years (up to and including April 2021) " in P2L70.
3) Why the patients had the ODI done? For the study o for routine clinical evaluation?
→ Thank you for pointing out the issue. In the past, the primary objective was to determine the probability of sleep apnea syndrome and evaluate it in dialysis patients at our institution. The current study is an attempt to determine the prognosis of these patients.
4) Is the ODI evaluated only one time for patients? Can the authors support (maybe with some literature) the fact that one ODI evaluation is enough to categorize patients?
→Thank you for pointing out an important issue. Severity of 3% ODI was evaluated with pulse oximetry only one night in this study. Patients who could not be adequately evaluated by overnight pulse oximetry were retested overnight pulse oximetry so that they could be evaluated again. However, no record of these measurements was kept, and it was unclear who took the measurements and how many times. Unfortunately, it was not known how many times the overnight pulse oximetry could be performed to accurately measure the severity of apnea. Thus, this measurement only one time was included the limitation of this study. I added in Discussion on P8L268-269 However, many studies evaluated the severity of sleep-disordered breathing with a tool, polysomnography, and/or overnight pulse oximetry, only one night.
5) Is the ODI evaluated in all the patients during the same inter-dialytic interval (long or short)?
→Thank you for pointing out an important issue. These patients in this study have received dialysis three times a week, i.e. same inter-dialytic interval. I added “All of the patients underwent hemodialysis with arteriovenous fistula or graft three a week.” in Results on P4L135.
6) It is useful to show the oxygen values measured during the ODI evaluation in the population and in the 4 subgroups.
→Thank you very much for pointing out an important issue. Saturation of pulse oximetry under resting conditions was added to Figure 1 and P4L152. While the median number of the saturation overnight was lowest in the severe group as expected. Thus, this saturation during overnight was as follows and was not added to Figure 1 as this result was expected. Unfortunately, mean SpO2 was not collected.
|
|
Total |
Normal |
Mild |
Moderate |
Severe |
p for trend |
|
|
N=134 |
n=22 |
n=52 |
n=37 |
n=23 |
|
|
overnight saturation |
96 (95-98) |
97 (96-98) |
97 (96-98) |
96 (95-97) |
95 (93-96) |
<0.001 |
7) The patients’ vascular access type (CVC vs AVF) is an important information to insert in the table.
→ Thank you very much for pointing out an important issue. All of these patients in this study underwent hemodialysis with AVF. I added “All of these patients underwent hemodialysis with arteriovenous fistula three a week.” in Results on P4L135. Thus, Table 1 did not show the type of vascular access.
We are very sorry. I misunderstood something and did not answer this question correctly. The patients in this study treated with hemodialysis, with not only AVFs, but also AVGs as shown below. However, there were no maintenance dialysis patients on dialysis with indwelling catheters.
Thus, I added below at Table 1, and changed " with the contralateral arm to the vascular access " at P3L90" and "All of these patients underwent hemodialysis with arteriovenous fistula or graft three a week." at P4L135-6,
|
Total |
0 |
1 |
2 |
3 |
p-value |
|
|
N=134 |
N=22 |
N=52 |
N=37 |
N=23 |
||
|
Vascular access |
|
|
|
|
|
|
|
AVF |
124 (92.5%) |
22 (100.0%) |
45 (86.5%) |
34 (91.9%) |
23 (100.0%) |
0.095 |
|
AVG |
10 (7.5%) |
0 (0.0%) |
7 (13.5%) |
3 (8.1%) |
0 (0.0%) |
|
|
CVC |
0 (0.0%) |
0 (0.0%) |
0 (0.0%) |
0 (0.0%) |
0 (0.0%) |
|

Reviewer 2 Report
The present study aims to “investigate the prognosis of patients undergoing hemodialysis, in terms of cardiovascular events and mortality, according to SDB severity determined by the 3% ODI with overnight pulse oximetry”.
The study has important limitations, the main ones underlined by the authors themselves. AHI was not investigated with polysomnography (PSG), which remains the gold standard for assessing the severity of sleep-disordered breathing, but only by overnight pulse oxymetry. Thus, even though Masuda et al reported in a letter to the editor of NDTPlus (ref 25) a significant correlation between AHI and 3% ODI in a workup screening for SDB in dialysis patients, in the same communication they also reported that the specificity of 3 % ODI for Sleep Apnea Syndrome (SAS) was limited to 55.2%. In fact, Masuda himself in a subsequent study attributed to pulse oximetry an inferior diagnostic accuracy for SDB (ref 21). This methodological limitation, together with the relatively small sample size and the observational retrospective nature of the present study preclude establishing a causal links between SDB and the prognosis. Therefore, it would be better to express the relationship between SDB severity and MACEs mortality as a "significant association" and SDB as a "risk factor", avoiding attributing a significant predictive or prognostic value to the reported results. Nonetheless, these results are very interesting and it would be important to confirm them through prospective randomized controlled trials.
The authors do not report some details about the pulse oximetry measurement. Was the pulse oximeter attached to the contralateral arm to the arteriovenous fistula (AVF) ? There are few and conflicting data about the influence of fistula on SaO2 measurement. Although Avitsian R et al did not find significant change in the SaO2 measured at the hand distal to the AVF as compared to the contralateral arm in a small group of hemodialysis patients, most studies report different results. For example, Modaghegh MHS evaluated SaO2 in 60 hemodialysis patients, 20 with mild hemodialysis access-induced distal ischemia (HAIDI) and 40 asymptomatic controls. He observed that, in patients with HAIDI, O2 Sat of the AVF side was significantly lower than in the contralateral side (92.9% ± 2.1% vs 95.6% ± 1.4%; P = .001) (MHS Modaghegh et al. Journal of vascular surgery 62: 135-142, 2015). Moreover, in a previous study, Lin G et al had evaluated 72 patients with a side-to-side primary AV fistula by pulse oximetry. SaO2 was measured before hemodialysis in both arms, the contralateral arm served as a control. They reported that the SaO2 differences between the hands of each patient before hemodialysis were 4% or more (Lin G. Am J Kidney Dis. 1997 Feb;29:230-2). Therefore, it is reasonable to think that the SaO2 measurement is significantly influenced by the presence of the fistula on the same arm, at least in a consistent group of patients. However, in the present study it not mentioned whether the measurement was taken on the arm contralateral to the fistula, nor whether and how many patients in each subgroup had HAIDI . The authors should address these important aspects in the methods.
Minor observations.
· In the lateral box of figure 1 has reported “5 patients had high age”; would be better "5 patients were over 85 years old"
· Other spelling errors should be corrected
Author Response
Reviewer2 comments
The present study aims to “investigate the prognosis of patients undergoing hemodialysis, in terms of cardiovascular events and mortality, according to SDB severity determined by the 3% ODI with overnight pulse oximetry”.
The study has important limitations, the main ones underlined by the authors themselves. AHI was not investigated with polysomnography (PSG), which remains the gold standard for assessing the severity of sleep-disordered breathing, but only by overnight pulse oxymetry. Thus, even though Masuda et al reported in a letter to the editor of NDTPlus (ref 25) a significant correlation between AHI and 3% ODI in a workup screening for SDB in dialysis patients, in the same communication they also reported that the specificity of 3 % ODI for Sleep Apnea Syndrome (SAS) was limited to 55.2%. In fact, Masuda himself in a subsequent study attributed to pulse oximetry an inferior diagnostic accuracy for SDB (ref 21). This methodological limitation, together with the relatively small sample size and the observational retrospective nature of the present study preclude establishing a causal links between SDB and the prognosis. Therefore, it would be better to express the relationship between SDB severity and MACEs mortality as a "significant association" and SDB as a "risk factor", avoiding attributing a significant predictive or prognostic value to the reported results. Nonetheless, these results are very interesting and it would be important to confirm them through prospective randomized controlled trials.
→Thank you for pointing out an important issue. As the reviewer commented, this study was a retrospective cohort study, thus we change to avoid to use predictive or prognostic words, and I changed “Impact of 3% oxygen desaturation index via overnight pulse oximetry on cardiovascular events and death in patients undergoing hemodialysis: a retrospective cohort study" in Title. And also changed “prediction” to “risk factor” in Abstract on P1L22, Discussion on P6L209, P7L211, P7L252-3, and Conclusion on P8L278-9 in red letters.
The authors do not report some details about the pulse oximetry measurement. Was the pulse oximeter attached to the contralateral arm to the arteriovenous fistula (AVF) ? There are few and conflicting data about the influence of fistula on SaO2 measurement. Although Avitsian R et al did not find significant change in the SaO2 measured at the hand distal to the AVF as compared to the contralateral arm in a small group of hemodialysis patients, most studies report different results. For example, Modaghegh MHS evaluated SaO2 in 60 hemodialysis patients, 20 with mild hemodialysis access-induced distal ischemia (HAIDI) and 40 asymptomatic controls. He observed that, in patients with HAIDI, O2 Sat of the AVF side was significantly lower than in the contralateral side (92.9% ± 2.1% vs 95.6% ± 1.4%; P = .001) (MHS Modaghegh et al. Journal of vascular surgery 62: 135-142, 2015). Moreover, in a previous study, Lin G et al had evaluated 72 patients with a side-to-side primary AV fistula by pulse oximetry. SaO2 was measured before hemodialysis in both arms, the contralateral arm served as a control. They reported that the SaO2 differences between the hands of each patient before hemodialysis were 4% or more (Lin G. Am J Kidney Dis. 1997 Feb;29:230-2). Therefore, it is reasonable to think that the SaO2 measurement is significantly influenced by the presence of the fistula on the same arm, at least in a consistent group of patients. However, in the present study it not mentioned whether the measurement was taken on the arm contralateral to the fistula, nor whether and how many patients in each subgroup had HAIDI . The authors should address these important aspects in the methods.
→Thank you for pointing out an important issue and for teaching detailed information. All of these patients in this study underwent hemodialysis with AVF or AVG and were evaluated for saturation oxygen with the contralateral arm to the vascular access. This was added in Methods on P3L90 In this study, we did not evaluate the percutaneous oxygen saturation in a finger on the same side of AVF.
Minor observations.
- In the lateral box of figure 1 has reported “5 patients had high age”; would be better "5 patients were over 85 years old" Other spelling errors should be corrected
→Thank you very much. I changed “5 patients had high age” to "5 patients were over 85 years old" in Figure 1. The spelling was checked and changed.

Round 2
Reviewer 1 Report
Dear authors. Thanks for reviewing. I think the article is now better than the first version.
Reviewer 2 Report
The authors appropriately responded to my comments